# Influence of Sizing Aging on the Strength and Fatigue Life of Composites Using a New Test Method and Tailored Fiber Pre-Treatment: A Comprehensive Analysis

Dennis Gibhardt * , Christina Buggisch , Lena Blume-Werry and Bodo Fiedler

Institute of Polymers and Composites, Hamburg University of Technology, Denickestraße 15,
21073 Hamburg, Germany
* Correspondence: dennis.gibhardt@tuhh.de; Tel.: +49-4287-88256

**Abstract:** Given the time-consuming and complex nature associated with the aging of composites, a novel fabric pre-aging method was developed and evaluated for static and fatigue testing. It allows for investigating sizing and interphase-related aging effects. This fast method is independent of the diffusion processes and the composites' thickness. Moreover, the new methodology offers enhanced analysis of the sizing, interphase, and fiber-related degradation of composites without aging them by conventional accelerated procedures or under severe maritime environments. For validation purposes, fiber bundle, longitudinal, and transverse tensile tests were performed with five different glass fiber inputs. Significant differences in the durability of composites were found for pre-aging and classical aging, respectively. The impacts of degradation of the single constituents on the fatigue life are identified by cyclic testing of untreated, pre-aged, and wet-aged composites. Here, it is evident that the interphase strength is likewise essential for the tension-tension fatigue performance of unidirectional composites, as is the fiber strength itself. In summary, the presented method provides industry and academia with an additional opportunity to examine the durability of different fibers, sizings, and composites for design purposes following a reasonable methodology.

**Keywords:** glass fibers; interphase; polymer-matrix composites (PMCs); pre-aging





## 1. Introduction

Knowledge of the long-term performance of fiber-reinforced polymers (FRPs), applied as lightweight materials for highly stressed components, is indispensable, as efficient and sustainable use requires a trouble-free service life over decades. Therefore, the fatigue behavior of FRPs is essential for numerous application areas. Consequently, industry and academia are especially interested in the lifetime behavior of all types of composites. In the maritime sector, durability under severe humid or wet environmental conditions is also of great importance, especially since glass fiber-reinforced polymers (GFRPs) are known to suffer under harsh conditions [1–6]. In this context, both the fatigue tests themselves and the additional effort dedicated to aging require a considerable amount of time. Consequently, several studies have been conducted on fatigue behavior under wet environmental conditions in the past [3,5,7–9]. However, compared to other fields, the available database for environmental fatigue is still not particularly large. Furthermore, some of the published results on environmental fatigue show contradictory results. While some authors found substantial lifetime decreases [1,3,7], others could hardly find any influence [10], and still others saw more load-level-dependent behavior [5,9]. Hence, it is difficult to clearly assign the reasons for the potential lifetime reduction, since the classical aging of composites in a water bath affects all constituents simultaneously.

Most studies on the effects of humidity- or water-related aging focus on polymeric matrix resins. The impacts on residual strength and stiffness [11–17], the glass transition temperature ($T_g$) [12,14,16,18,19], and the water absorption process in general [1,12,13,16,20,21]

are frequently investigated. Therefore, it is known that the absorbed water interacts in a complex manner with the resin's molecular structure. While strength, stiffness, and $T_g$ are regularly reduced by the plasticizing effect, significant physical aging occurs, especially with accelerated aging at elevated temperatures [12,14].

Glass fibers are susceptible to hydrolytic attacks leading to slow degradation by leaching of surface ions [22,23]. As intrinsic flaws are exposed and enlarged, the resulting fiber strength decreases with time [24,25]. However, to what extent the fibers are protected from the water by the matrix and the interphase region within a composite is a topic of frequent discussion. The consensus in this context is that the integrity of the fiber/matrix interphase will mainly contribute to sufficient protection. Due to its microscale size, investigating aging effects on the fiber/matrix interphase is challenging. With atomic force microscopy investigations, it was revealed that the interphase properties are severely reduced by water absorption, with the weakened region significantly increasing in size [26,27]. Most commercial glass fiber sizings consist mainly of reactive organofunctional silanes as coupling agents and matrix resin-compatible film formers [22,28]. Additionally, sizings often contain antistatic agents, lubricating agents and surfactants, which make sizings a highly complex and difficult component to explore [28]. Since most of these ingredients are reactive, they are subject to natural aging, which results in declining functionality of the sizing over time [29]. However, as the exact composition of the sizings is usually not known, there exists little knowledge about the aging behavior of sizing. Most laboratory studies have tried to reduce the number of sizing ingredients to mainly silanes and film formers to reduce the complexity [30–32].

The important studies and findings about the aging of fiber sizings by Plonka et al. [32] and especially Peters [33] or for fibers by Brown et al. [22] were used as a starting point for developing and extending a tailored fiber sizing aging methodology. Although the idea of aging fibers and sizings is not entirely new, the implementation and analysis of the effects on the static and fatigue properties of the resulting composites is not yet reported. To complement and improve the testing and prediction opportunities, the authors recently published findings on the potential of a novel fiber pre-aging procedure [34] to compare the sizing and interphase durability of GFRPs. The main advantage of this fiber-aging-based methodology is the possibility of directly aging individual composite's constituents. For this purpose, the fibers are stored for several weeks under high humidity and temperature and subsequently infused with resin. Therefore, aging becomes independent of diffusion processes and results in considerable time savings compared to conventional methods. Within the framework of this research work, the newly developed method is further evaluated and applied to fatigue loadings. Furthermore, it is extended to wet pre-aging in a water bath to additionally characterize the impact of fiber degradation.

In order to examine the effects on different fiber types, a total of five different fiber inputs (different non-crimp fabrics (NCFs)) were used for this study. To reveal the durability of the fiber, the long-term behavior of the fibers in contact with high humidity or water was examined using fiber bundle tests and single fiber fragmentation (SFF) tests. For fatigue testing, the pre-aging approach provides a novel possibility to highlight the individual impacts of a decreased interphase, fiber and interphase, or matrix strength on the lifetime behavior. In this context, it is demonstrated that the interphase properties are essential for the durability of unidirectional (UD) composites. Furthermore, the results presented support the micromechanical models recently evolved by Sörensen et al. [35,36], Castro et al. [37] and by Fazlali et al. [38], which all describe the damage growth process in UD composites under fatigue loading with a specific focus on the fiber/matrix interphase. Thus, the new test methodology provides an opportunity to experimentally verify the micromechanical models. By employing this methodology in the industry, fast and straightforward comparisons of sizing and fiber durability can be achieved to estimate maritime use capabilities.

## 2. Materials and Methods

### 2.1. Design of Experiments and Theoretical Background

By investigating composites' durability, the typical procedures are based on the storage of the material in hot/wet conditions or even completely immersed in water. Unfortunately, these types of accelerated aging methodologies are barely standardized, and the results must always be considered in the context of the actual operating environment. As absorbed water acts simultaneously on all constituents of the composite, it is unclear whether the respective degradation processes occur at the same rate and duration. As a consequence, it is difficult to attribute the changes in mechanical properties to the specific influence of fiber, matrix, and interphase.

For the composite industry and academia, the degradation stability of fiber sizings is of particularly high importance, as it has to be known within which time frame fibers can be further processed into composites without drastic performance reductions. From a chemical point of view, several mechanisms are involved in the aging of sizing and composite interphases. First, the hydrolysis reactions of the organofunctional silanes in the fiber/sizing interphase under the impact of moisture and heat are expected to reduce the interfacial adhesion, since the condensation reaction of the Si-O-Si bonds is reversible [33]. Here, water molecules adsorb to the surface, diffuse into the sizing and incrementally resolve the fiber/sizing bonding [29,39]. Of course, this is especially true for non-processed fibers, where moisture can more easily reach the fiber/sizing interface. Nevertheless, these chemical bondings are theoretically still susceptible to degradation even in the fiber/matrix interphase formed and could represent a link between classical aging in the composite and sizing aging. Furthermore, hydrolysis of chemical functions, either in silanes or polymeric film formers, occurs in the presence of moisture and is accelerated with heat [32,33]. Although it is not clear to what extent the film formers will be integrated by chemical bonding into the matrix [28], it was found by FTIR investigations that the reactivity of, e.g., epoxy functions, decreases during storage and aging [33,40]. The reduced reactivity of the sizing then also causes a significant reduction in fiber/matrix adhesion in the subsequently manufactured composites. However, besides these frequently referred processes, a number of unknown ingredients, such as lubricants, are part of commercial sizings. The complex interplay between the components responsible for adhesion and those required for process technology certainly also affect the resistance and long-term durability of the sizings, e.g., by changing the morphology, accessibility, or pH value of the interphase. Therefore, a standardized methodology for durability comparison is needed.

To address this challenge, a pre-aging method was designed that allows the targeted aging of only the individual constituents of the composite by aging of the NCFs before composite manufacturing. This makes it possible to investigate the aging behavior of fabrics during transport and storage up to production and to measure the effects of selectively aged constituents on the composite properties. In particular, the aim here is to estimate the impacts of aged fiber sizings and fiber-matrix interphases on the mechanical properties and fatigue life of composites. The studies of Peters [33] and more recently of Cech et al. [29] suggest that the aging of sizings under conventional environmental conditions is a serious process, whose rate depends directly on the sizing formulation. For 11-year aging under production hall conditions, the sizing aging effects reduced the composites' interfacial strength during the first 40 months [29]. Afterward, no further reductions were found during the following seven years. Furthermore, considering the results of Peters [33] and Gibhardt [34], it seems likely that there is a lower limit of remaining fiber/matrix bonding capability for each sizing. Questions surrounding the rate of the aging progress and whether this is a linear process in time cannot be answered directly on the basis of the results published to date.

For the development of a defined fiber and sizing pre-aging methodology, it is assumed that the accelerating effect of the temperature follows an Arrhenius relationship since most chemical reactions as well as diffusion and aging follow this principle [3,17]. However, it is more difficult to estimate the accelerating effect of increased humidity, since its presence

in the natural aging process is a basic requirement that is not always equally prevalent. Studies conducted by LeGuen Geffroy et al. [14] on the physical aging of epoxy resins show a dramatic acceleration (factor of ten) of aging processes by the presence of water in this context. Based on the previous assumptions and with the aim of not generating physical or chemical effects that would not occur under normal conditions during natural aging of up to 40 months, the aging process was carried out at 50 °C and 80% RH initially for one, five, and ten weeks. As described in more detail in Section 3.3, it was found that the aging process seemed to be mainly completed within five to ten weeks under these conditions. This corresponds to an accelerating factor of about 16–32 and seems reasonable with regard to the Arrhenius principle for a temperature increase of about 30 °C under significantly increased humidity. Additionally, the fabric pre-aging is implemented in two ways, allowing either to age the fiber sizing only or the sizing and the fiber simultaneously. The developed aging methodology is schematically shown in Figure 1. Following the time-dependent pre-tests, the fabrics are stored either at 50 °C and 80% RH for five weeks or in a water bath at 50 °C for the same period in case of all main investigations. As both pre-aging processes are applied before resin infusion, they do not affect the composite matrix. Thus, the effects of matrix aging can be excluded from later test results.

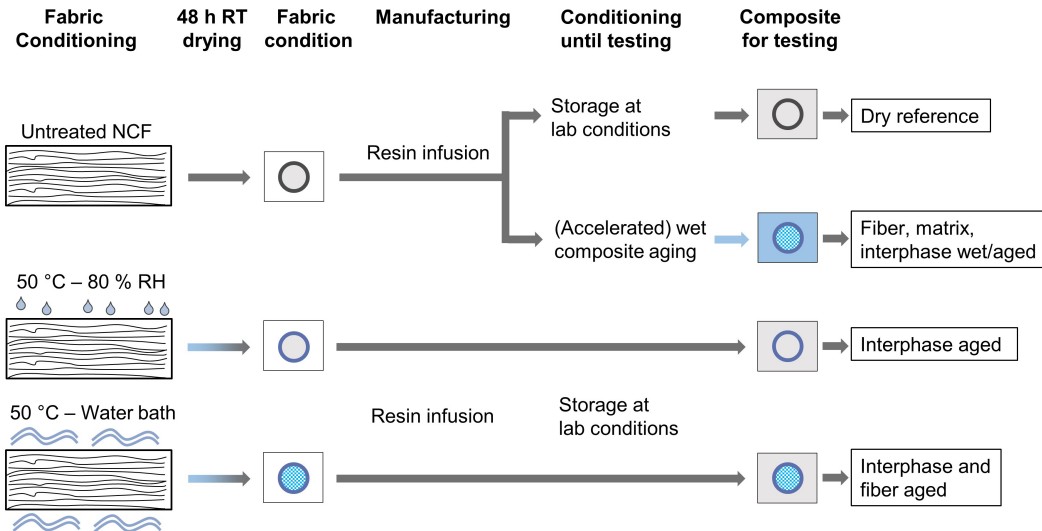

**Figure 1.** Schematic presentation of classical composite aging and new pre-aging methodology.

Fiber bundle and longitudinal tensile tests on composites examine how both pre-agings affect fiber strength and fiber-dominated properties. The development of the interphase strength is investigated by means of transverse tensile tests. In addition, SFF tests were performed to investigate the effects of aging on a micromechanical scale. Impacts on the lifetime of the composite materials are analyzed using tension-tension fatigue tests. For comparison, all composites were also conventionally aged in a water bath at 50 °C. All quasi-static test results are summarized in Table A1 in the Appendix A.

## 2.2. Materials, Conditioning, Manufacturing, and Specimen Preparation

In total, the study consists of comparing the effects of aging with five different fiber inputs to manufacture UD GFRP composites. In order to not emphasize any material or manufacturer, the non-crimp fabrics are neutrally designated as "Fiber A–E" based on the different fibers and sizings. All NCFs were processed into unidirectional laminates with 0° fiber orientation and a target fiber volume content of 50%. As the areal weights of the NCFs varied between 600 $\frac{g}{m^2}$ and 1192 $\frac{g}{m^2}$ the number of individual layers was adjusted. The resulting fiber volume fractions were between 45.0% and 49.7%. Consequently, the longitudinal strength was normalized to a fiber volume fraction of 50%. The amount of backing fibers in the transverse direction of the NCFs was small ($\leq$5.0%) in all cases, with the exception of fiber C, where it was about 10%. Consequently, the transverse strength was taken

here from the knee of the stress–strain diagrams and normalized to the stress acting in the 90°-layers, as the backing fibers take a proportion of the load in fiber direction during transverse testing. Therefore, the stress distribution between the different orientations is micromechanically calculated based on the fiber proportions.

The NCFs were infused with the two-component epoxy resin system Hexion (now Westlake) EPIKOTE™ Resin MGS™ RIMR 135 and the amine hardener EPIKURE™ Curing Agent MGS™ RIMH 137 that is widely used in wind turbine blade manufacturing. An automatic saw with water-cooled aluminum oxide blades was used to prepare rectangular specimens for static tests. Dogbone-shaped fatigue specimens were prepared using a CNC mill equipped with diamond bits. After manufacture, a subsequent check for uniform quality by visual inspection confirmed that there were no pores or defects. The cutting edges of the samples were polished before fatigue testing in three steps with sandpaper of grid sizes up to 2500 to avoid cutting edge effects. The fiber volume content $\varphi_f$ was determined by burn-off testing. All pre-aged and untreated dry reference samples were dried in a vacuum oven at 40 °C for 48 h before testing.

For humid pre-aging of NCFs, a climate chamber (CTC256, Memmert, Germany) and for pre-aging in distilled water, a temperature-controlled water bath was used. After five weeks of pre-aging, all fabrics were dried under laboratory conditions for 48 h. Subsequently, a resin transfer moulding (RTM) process at 50 °C for 10 h, including a 2 bar over-pressure, was used for composite manufacturing. Post-curing followed at 80 °C for 16 h. The SFF samples were manufactured using silicone molds. Here, individual fibers are clamped in the mold, prestressed with a weight of 8.5 g, and cast in epoxy. After curing and post-curing, the specimens were polished and a quasi-static tensile test was performed. The tests were stopped before the final failure of the specimens, but after the yield zone was reached. Subsequently, the specimens were analyzed using polarized light under a VHX 6000 microscope (Keyence, Japan).

### 2.3. Mechanical Testing

Static tensile and transverse tensile tests were performed following the DIN EN ISO 527-4 standard. Rectangular specimens of 250 mm length, 15 mm width for longitudinal, and 25 mm width for transverse tests were tested using universal testing machines (Z100 and Z10, Zwick-Roell, Ulm, Germany) with mechanical extensometers and 100 kN or 10 kN load cells. Fiber bundle tests were performed according to DIN EN 1007-5 standard, using a universal testing machine (Z2.5 or Z10, Zwick-Roell, Ulm, Germany) with 2.5 kN or 10.0 kN load cell and strain measurement via contactless video extensometer or crossbar movement, respectively. The fiber bundles were 260 mm long (200 mm gauge length) and adhered with superglue to sandpaper at their gripping edges. This allowed a uniform load introduction. The fiber failure probability $P_j$ and tow strength $\sigma_{tow}$ were evaluated according to the standard. The fatigue tests were performed using a universal hydraulic testing machine (Instron, Norwood, MA, USA) equipped with a 100 kN load cell. Only 2 mm thick UD dogbone specimens were investigated in tension-tension fatigue tests (0°-direction) with a load ratio of R = 0.1 at a frequency of 3 Hz. For the evaluation of the SFF tests, the interphase shear strength ($\tau_s$) was calculated according to [41]:

$$\tau_s = \frac{\sigma_f \cdot d_f}{2l_c},\tag{1}$$

with the fiber strength $\sigma_f$, fiber diameter $d_f$, and critical fiber length $l_c$, calculated as the average of the double fiber break distances of each of the five smallest fragment pieces.

### 2.4. Fatigue Specimen Design

As rectangular specimens according to DIN EN ISO 527-5 standard regularly fail inside the gripping region or by longitudinal splitting, a dogbone-shaped design was implemented for fatigue testing of UD composite specimens. The development of the exact sample geometry was initially based on dogbone variations from previous studies [42,43].

In contrast to these studies, a limitation was established that a total sample length of 250 mm should not be exceeded.

Finite element analysis (FEA) was used to achieve a structured improvement of the specimen geometry. In detail, the width of the specimen in the free (parallel) length $W_p$ and at the edges $W_e$, as well as the length of the parallel section $L_p$ were systematically varied. The scheme of the variations and the parameters of the final geometry are shown in Figure 2. For analysis, the material model Composite Damage (CompDam) Progressive Damage Analysis Software of the NASA Langley Research Center Hampton, VA, USA, was used [44]. This code is a material model of fracture mechanics in the field of continuum damage mechanics (CDM) and is intended for use with the Abaqus FEA program from Dassault Systems. The software describes the damage mechanisms in the composite by means of an FE model in which each layer is represented. Matrix crack kinematics are represented according to the deformation gradient decomposition approach [45]. The fiber damage due to tensile stress is described by conventional strain softening in the CDM. Specimens are simulated in Abaqus FEA as quasi-static tensile test with an explicit solver. Here, the specimen is fixed on one side and loaded on a reference point on the other side. The mesh consists of 1 mm C3D8R elements starting from the clamping and oriented along the radius. All parameters required for the GFRP material are selected based on the results of own tests or comparable studies [42,46]. For the evaluation of the FE analysis, the failure index matrix (FIM) is analyzed because splitting cracks initiate the final failure in the UD laminate. An FIM value of ≥1 displays a matrix crack that has fully formed. To evaluate geometries, all specimens were simulated at a nominal stress of 450 MPa (0.5 UTS) in the free (parallel) test length and the FIM value was compared in the critical edge region. Based on the simulation results, three geometries were selected for fatigue testing (based on fiber D). The best-performing shape (lowest longitudinal splitting, no splitting from the edges, highest fatigue life) was then chosen for the main investigation. At the selected comparative load level, the desired geometry endured in average 230.000 load cycles (fiber D), which is about 13.9 times more than the rectangular benchmark geometry. The corresponding fatigue results are shown in Figure 3.

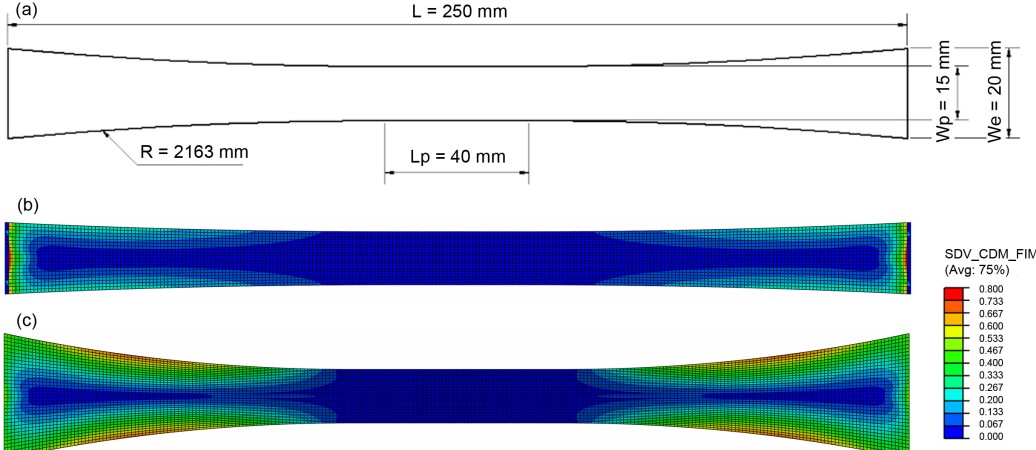

**Figure 2.** Schematic presentation of the final dogbone geometry (**a**). $W_e$, $W_p$ and $L_p$ were varied during the FEA investigation, as shown in (**b**,**c**).

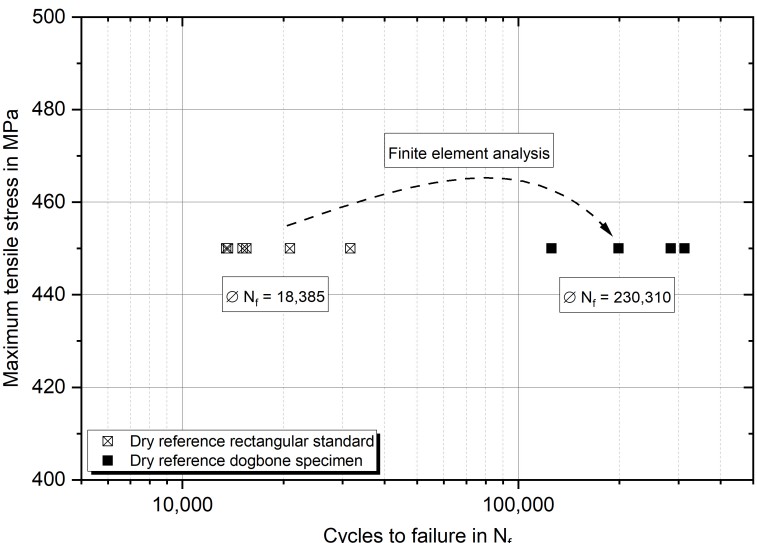

**Figure 3.** Results of the fatigue test of the geometry of the developed dogbone specimen compared to standard rectangular specimens at a maximum load level of 450 MPa.

## 3. Results and Discussion

### 3.1. Impact of Material Selection on Static Properties of UD Composites

Among the most important questions manufacturers have to address include which materials to use and how the choice will affect performance. Since the market for glass fibers and resins is very large, the combination opportunities are huge. Although a single study can hardly cover this market, it is possible to examine basic similarities and differences using a smaller data set. The tensile properties of each fiber-type and fiber-matrix combination investigated are evaluated as benchmark values. Emphasis was also placed on processing the NCFs into the composite as soon as possible after fabrication to minimize aging effects due to storage. The bundle strength $\sigma_{tow}$, the composite strength in the fiber direction $\sigma_{0°}$, and the transverse strength $\sigma_{90°}$ are shown in Figure 4 for the initial dry reference conditions. Here, the tensile strength in fiber direction is normalized to a $0°$-fiber volume fraction of 50% in order to achieve a proper comparison.

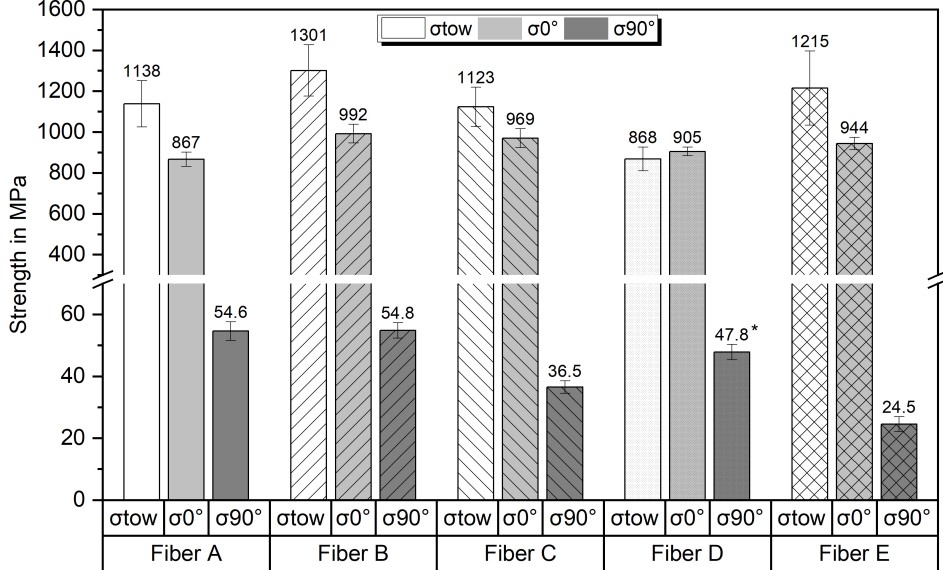

**Figure 4.** Quasi-static absolute fiber bundle, longitudinal and transverse tensile strength of UD GFRP composites with different NCF input (initial dry reference condition). Shades are corresponding to the NCF input. *: Value according to [47].

When observing the fiber bundle strength, it is noticeable that there are differences between the systems considered. The range of average strengths is between 1123 MPa and 1301 MPa, which means a variation of up to 14%. However, as the standard deviations are typically high for this type of test method, the differences are not as significant as the numbers would lead one to expect. The lowest strength measured by far (fiber D) can be explained by the fact that the fiber bundles could probably not be completely detached from the fabric without damage. In this case, the fiber bundles were noticeably stitched together. Compared to the average fiber strength usually expected from single fiber tests, the strength measured with the bundle test is clearly only about half as high. The reasons for this are mainly friction and misalignment of the fibers during the test. However, it has already been shown that the methodology is well suited to detect relative variation within a series of studies [24].

The fact that the composite strength in the fiber direction of fiber D is in the same range as that of all other systems also indicates that the fibers themselves are not weaker. The strengths of the laminates in the fiber direction cover a range from 867 MPa to 992 MPa, whereas the highest laminate strength was reached by fiber B, which showed the highest bundle strength as well. The difference between the lowest and highest strength is about 13% and thus relatively large. This can be due to different fiber and interphase strengths and due to the NCF structures. High strength in the direction of the fibers requires effective load transfer and the lowest possible influence of binder and backing fibers.

The most significant differences can be found in the transverse tensile strength. Here, the highest dry reference strength of the fiber B composite (54.8 MPa) is about 2.2 times higher than the lowest strength of the fiber E composite (24.5 MPa). Because the transverse tensile strength of UD composites is regularly taken as an indicator for the interphase quality, it is directly influenced by the fiber sizing and its interaction with the polymeric matrix. A low transverse tensile strength is critical for the initiation of damage and progress in composite structures, as it directly contributes to the lifetime. Therefore, a high transverse tensile strength is typically required as a quality characteristic and should be regularly verified as part of an incoming material inspection.

Since, the transverse tensile strength also depends on the fiber/matrix bonding and the matrix properties, this factor was investigated separately. In Figure 5, the transverse tensile strengths of the fiber systems A–C are presented for three different but relatively similar epoxy resins (based on amine hardeners) used as matrix material. Here, resin one is the RIMR135/H137 benchmark system, and the others are applied in rotor blade manufacturing as well. While the differences caused by the different epoxies of systems A and B are considerable with up to 13%, system C proves to be even more sensitive to the matrix choice. The resin-dependent strength variation was up to 15.8 MPa (38%) in this case. In general, matrix system three resulted in the lowest strength throughout the tests. Thus, it can be assumed that the resin choice may well have a significant influence on the fiber/matrix bonding and the resulting mechanical properties, even if the sizings and resin types are fundamentally matched.

### 3.2. Accelerated Aging of GFRP Composites

The classic aging of UD GFRP composites in 50 °C water reveals strong variations in the durability of the main mechanical properties depending on the fiber input, as shown in Figure 6. The smallest differences were found after the five-week water bath aging period for the bundle strength. With strength reductions ranging between 16–20%, these are all in a similar range and even perform almost 10% better than the comparative results reported in [48]. The wet-aged composites were tested after ten weeks in the water bath when apparent saturation was reached. Here, the reductions in 0°-composite tensile strengths ranging from 22% (fiber B) to 53% (fiber C) vary significantly more than the bundle strengths and cover approximately the entire range previously reported in other studies of accelerated aging with GF/EP composites [4,5,49,50]. In contrast to the literature comparison, in the present case, the matrix resin system is always the same. Therefore, the differences in

durability can be mainly attributed to the fibers and the fiber/matrix interphases. Given that the 0°-strength is largely determined by the fibers, it can be assumed that these have a correspondingly high contribution to the overall reduction. Decreased interphase strength is more associated with earlier and accelerated damage progression and reduced load transfer. However, nevertheless, taking the bundle strength results into account indicates that the aging process in the composite is certainly also affected by the interphase and the fiber/matrix bonding. For fibers A and B, the longitudinal tensile strength of the composites decreases by nearly 25% and 22%, which is approximately 6–8% more than the bundle strength. However, in comparison, this is the smallest reduction. For fiber systems C and D, the strength drops by 53% and 41%, respectively. With residual strengths of around 500 MPa or less, these composites are already enormously weakened.

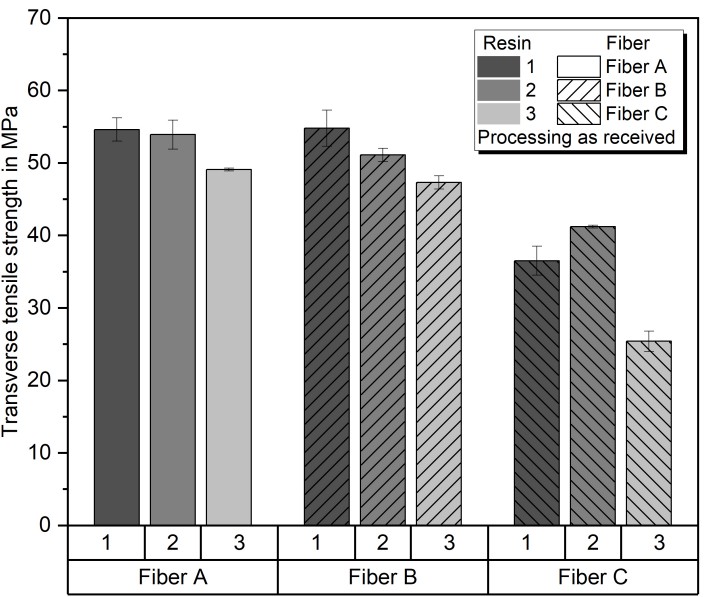

**Figure 5.** Transverse tensile strength of fibers A–C with regard to the use of different epoxy resins (with amine hardeners) as matrix materials.

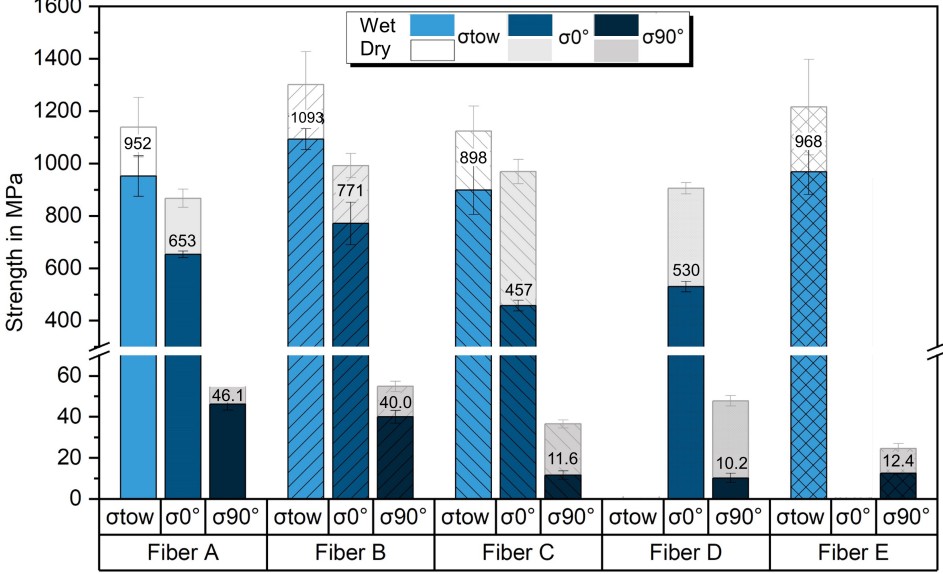

**Figure 6.** Quasi-static absolute fiber bundle, longitudinal and transverse tensile strength of UD GFRP composites with different NCF input after accelerated aging of the composites in 50 °C water until saturation. Dry reference values are shown in grey shade for comparison. Patterns are according to the NCF input.

In this context, it is particularly interesting to also peer into the development of the transverse tensile strength. Similarly to the 0°-strength, systems A and B perform significantly better than the others. Even after ten weeks in 50 °C water, the 90°-strength is with 46 MPa and 40 MPa still higher or as high as the dry strength of the comparative systems. Residual strengths of around 10 MPa, on the other hand, are very low and critical for any application. On the basis of the present results, it can be assumed that the durability of the fiber sizing against moisture and temperature in particular is decisive for the overall resistance of the composite. In addition, a durable interphase also seems to significantly better protect the fibers themselves and consequently maintain the 0°-strength better. However, it must be pointed out that accelerated aging at elevated temperatures produces a condition that does not always occur in real applications. The fact that the resistance to wet aging under low temperatures may be better depending on the fiber system was recently shown in [51]. Consequently, the aging behavior of NCFs and composites is considered via alternative approaches in the following.

### 3.3. Standard and Tailored Aging of Sizing and Fibers

Due to the chemical nature of the fiber sizing, its reactivity typically reduces over time. The main reasons for this are the hydrolysis reactions taking place, especially when reactive groups are exposed to moisture and heat. For the specific fiber systems investigated, these processes are mainly related to the chemical functions of reactive epoxy groups, which are the main components of film formers and silanes. Therefore, they make up a large part of the total sizing. The hydrolysis of epoxy functions was demonstrated by Peters et al. [33] with FTIR analysis of sizings and more specifically the evolution of the epoxy peak at 912 cm$^{-1}$. Based on this knowledge, the long-term development of the NCF (sizing) performance was investigated for storage under laboratory conditions or elevated temperature and moisture conditioning by means of transverse tensile strength tests. As shown in Figure 7, the transverse strength in dry conditions decreases over standard NCF storage time for both investigated fiber types. However, while the five-month reduction was relatively similar, the long-term performance was found to be dramatically different. While the transverse strength of a laminate produced two years after NCF production in the case of fiber A still shows a relatively high strength of about 44 MPa, the strength of the fiber D laminate was only about 18 MPa.

In comparison to the standard (natural) aging process, the results gained from the accelerated and tailored NCF pre-aging tests of up to ten weeks are shown as well. Here, two aspects become clear at first. On the one hand, aging can be dramatically accelerated (by a factor of four–eight) and, on the other hand, it is shown that after five to ten weeks almost the same residual strengths are achieved as after long-term standard aging. This means that the tailored aging process does not seem to promote other processes that would not occur under natural conditions. Furthermore, in this short time, a constant and individual level of the lowest transverse strength is reached for each fiber type, which is in good accordance with the results presented by Cech et al. [29]. Aging at elevated temperatures and moisture for even longer times does not further reduce the properties. Therefore, the process is suitable for investigating and comparing the sizing stability of different systems qualitatively and quantitatively within a few weeks.

Apart from storage stability, the question naturally arises whether the new pre-aging approach can also be used to make any statements about the long-term performance of the resulting GFRP laminates. Therefore, the residual transverse strength after composite aging in a 50 °C water bath (WB) and the residual transverse strength after five-week NCF pre-aging in 50 °C humid air (Pre HA) are compared in Figure 8. From the results obtained with the five fiber systems under investigation, a clear correlation between NCF pre-aging durability and composite water bath aging appears. The transverse tensile strength after NCF humid pre-aging was comparably low for the same three fiber systems that showed the largest reductions during composite aging as well. Even if the correlation found does not necessarily apply to all other glass fiber systems on the market, this result is remarkable.

Although a chemically bonded and fully formed fiber/matrix interphase should have a constitution different from pure sizing, the effects of moisture and temperature are clearly comparable. One reason for this could be that some of the weaker fiber/matrix interphases might undergo hydrolysis reactions during water bath aging, which results in a similar detachment of the interphase compared to the non-formation of bonds due to the aging of the sizing in advance. In other words, a protective function of the sizing also seems to act in the composite if the sizing itself has a high ability to withstand hygrothermal aging. This could also be an explanation for why the mechanical properties of the different composites vary considerably after classical aging in a water bath (Figure 6). Again, all GFRP composites with a low transverse tensile strength after aging (and pre-aging) show a significant reduction in 0°-tensile strength as well. A conclusive interpretation could be that the interphase remains protective for the fiber when the sizing has a high level of resistance to degradation. However, on the basis of the present study, it appears quite possible to distinguish more resistant systems from less resistant ones using the new NCF aging method.

To determine the effect of the presented aging methodology on the interphase properties in more detail, micromechanical experiments in the sense of SFF tests were performed. Therefore, single fibers embedded in epoxy resin were tested in tension until fiber break saturation sets in. Afterwards, the specimens were analyzed under polarized light. On the one hand, this makes it possible to measure the number of fiber breaks and their distances, and on the other hand, it allows the determination of the extent of fiber/matrix debonding. Representative sections of fiber breaks of the dry, humid, and wet pre-aged fibers are shown in Figure 9. While the mean break distance for the dry fibers is $550 \pm 67$ μm, which corresponds to an interphase shear strength of 31.8 MPa using Equation (1), it is clear that the humid pre-aged fibers show a significantly larger distance. Here, the fiber fracture distance of $1904 \pm 479$ μm equals an interphase shear strength of only about 7.7 MPa. It is also noticeable that the fiber/matrix debonding in the aged case is many times greater than in the reference state. Consequently, the applied load cannot be introduced into the fiber, leading to strains and stresses in the matrix. This can also be seen in the cross-wise strain peaks in the stress-optical images. A special case is the pre-aging of the fibers in a water bath. Contrary to expectations, the fiber break distance is reduced in this case, which would correspond to an increase in the interphase shear strength. However, two important factors must be taken into account here: Unlike aging in humid air, the fiber itself is weakened by direct contact with water. Additionally, the fiber/matrix debonding is again much more pronounced compared to the reference. Following this, both the interphase and the fiber strength are significantly reduced (which is in line with the quasi-static test results shown in the Appendix A in Table A1 for NCF pre-aging in a water bath). To approximate the effect in numbers, it could be assumed, for example, that the interphase strength is reduced to a similar extent as in the humid case. Then, following Equation (1) the residual fiber strength after water bath aging would be about 500 MPa, which is less than a quarter of the original strength. On the other hand, the interphase strength could also be somewhat higher than in the humid aging case, since the sizing in the initial application is also condensed from an aqueous solution onto the fiber surface. Nevertheless, based on the micromechanical tests, it can be said with certainty that, compared with the reference, both the fiber and the interphase strength decrease significantly. By using the new pre-aging approach, it is now possible to investigate the effects of a reduced interphase strength independently or in combination with a reduced fiber strength on the properties of a composite.

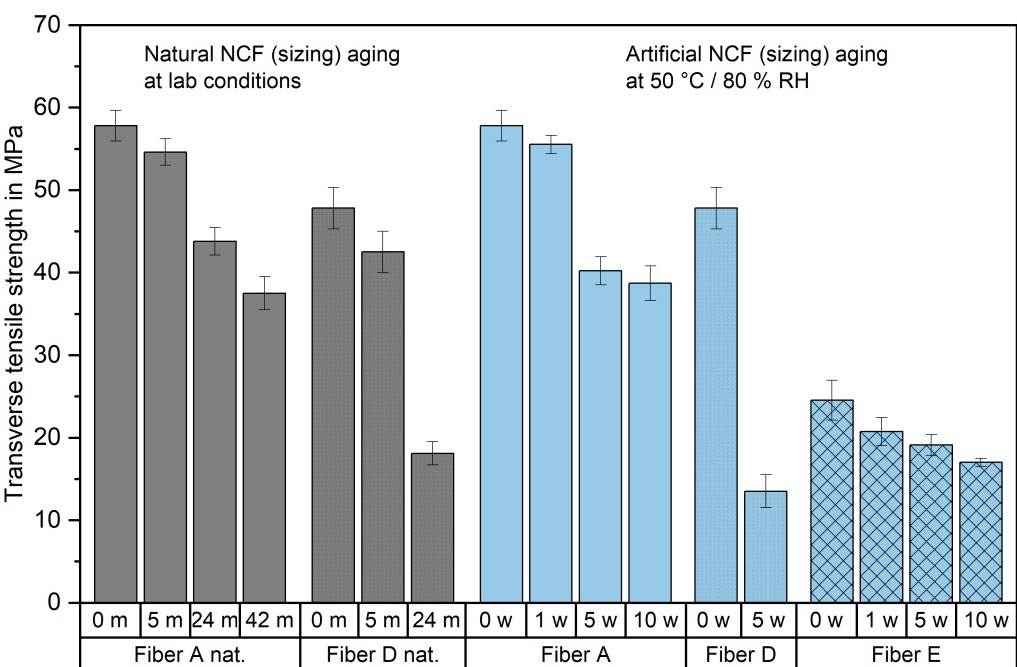

**Figure 7.** Development of transverse tensile strength due to standard NCF sizing aging under laboratory conditions (**left**) and tailored NCF aging at 50 °C and 80% RH (**right**). Grey shades represent natural aging, blue shades accelerated aging. The duration of standard aging is given in months (m), and tailored aging is given in weeks (w).

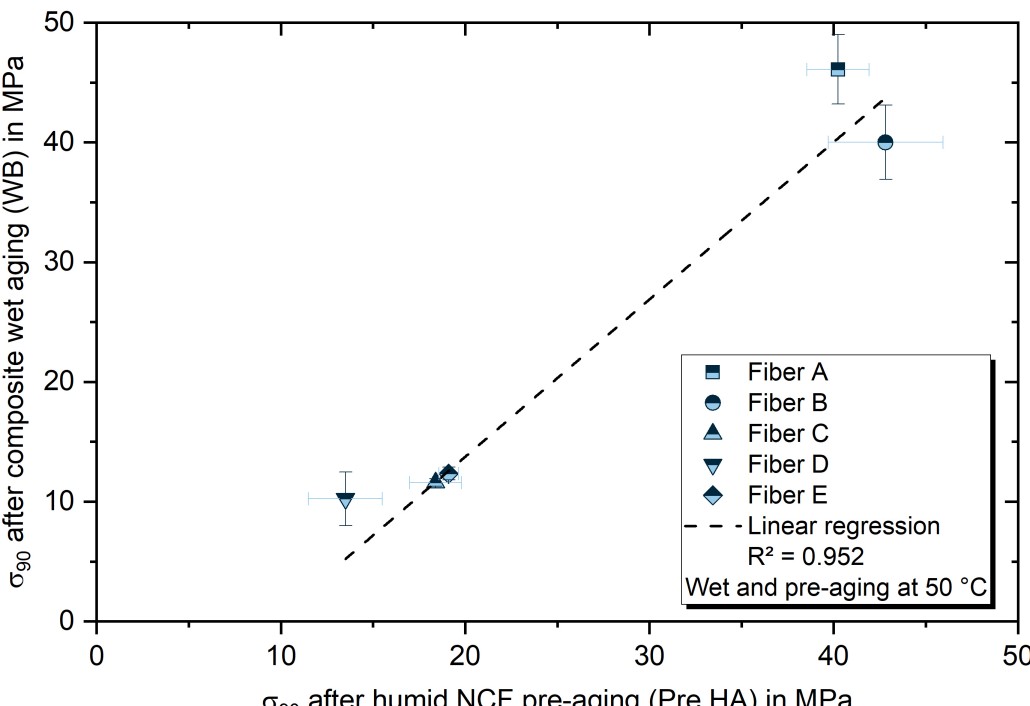

**Figure 8.** Correlation between transverse tensile strength after composite wet aging (WB) and NCF humid pre-aging (Pre Ha) for all investigated NCFs.

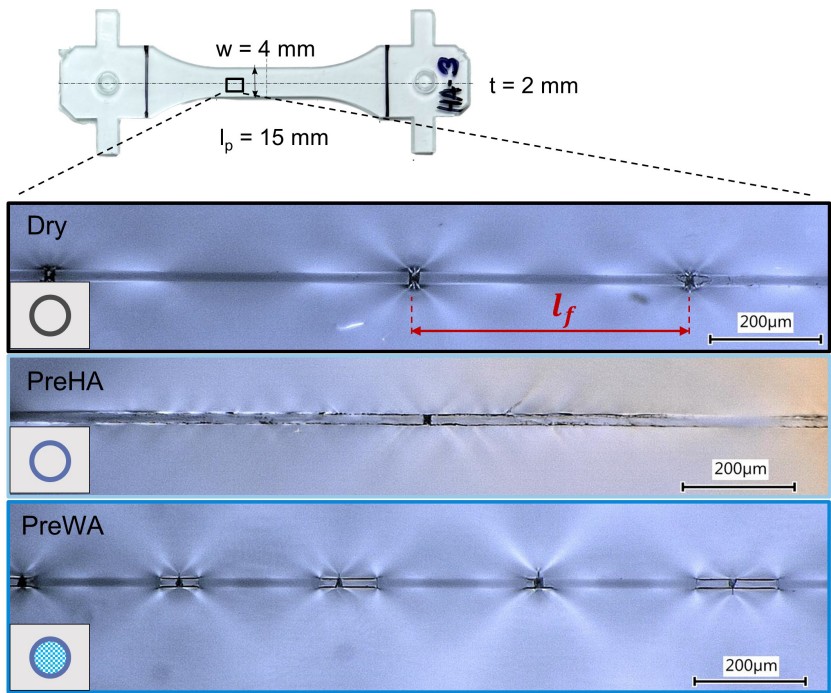

**Figure 9.** Single fiber fragmentation test specimen (top) and representative microscopic pictures of fiber breaks for the different aging conditions.

### *3.4. Impact of Aging and Material Selection on Fatigue Life*

Following a tension-tension fatigue study based on three of the fiber systems and different aging modes, the performance is presented to evaluate how severe properties of the interphase affect the fatigue life of unidirectional 0°-composites. The Wöhler-curves of the dry reference specimens are shown in Figure 10 for comparison of the basic systems. Interestingly, clear differences can already be observed here. The fatigue performance of fiber system A is significantly better than that of systems D and E, although the fiber volume fractions and NCF fabric structures are very similar. While the slopes of the curves are quite similar and in accordance with other studies such as those published by Mandell et al. [52], the lifetime achieved can differ by up to a decade per load level. If the results are related to the transverse tensile strengths (Figure 4), a correlation can also be seen here. The lower the transverse tensile strength of the systems, the poorer their fatigue performance.

The fatigue results for humid pre-aged, wet pre-aged, and in the case of fiber system D, also composite wet-aged specimens are shown in Figure 11. By comparing the individual results and Wöhler-curves, several distinctions can be observed. First, the reduction in interphase strength due to humid pre-aging leads to a significant reduction in fatigue life without changing the slope of the Wöhler-curve. The lifetime reduction can be up to a decade at the same load or 100 MPa at the same lifetime. In terms of absolute load level, this corresponds to a reduction of tolerable fatigue loads by up to 20%, which is particularly critical, since the reduced strength is not observable in the quality of the composites.

The wet-aging of the NCFs and the associated reduction of interphase and fiber strength prior to composite fabrication further reduce fatigue performance for both systems. In addition to the pure reduction in lifetime, there is also a decline in the slope of the Wöhler-curve. The change in slope is due to reduced fiber strength. At high loads, disproportionately more fibers fail than at low loads, directly reducing the load-bearing capacity and lifespan of the specimen. At lower loads, in contrast, the load transfer between the fibers is again of high importance. Due to the weakened interphase, this is inferior to the reference condition but comparable to the humid pre-aging case.

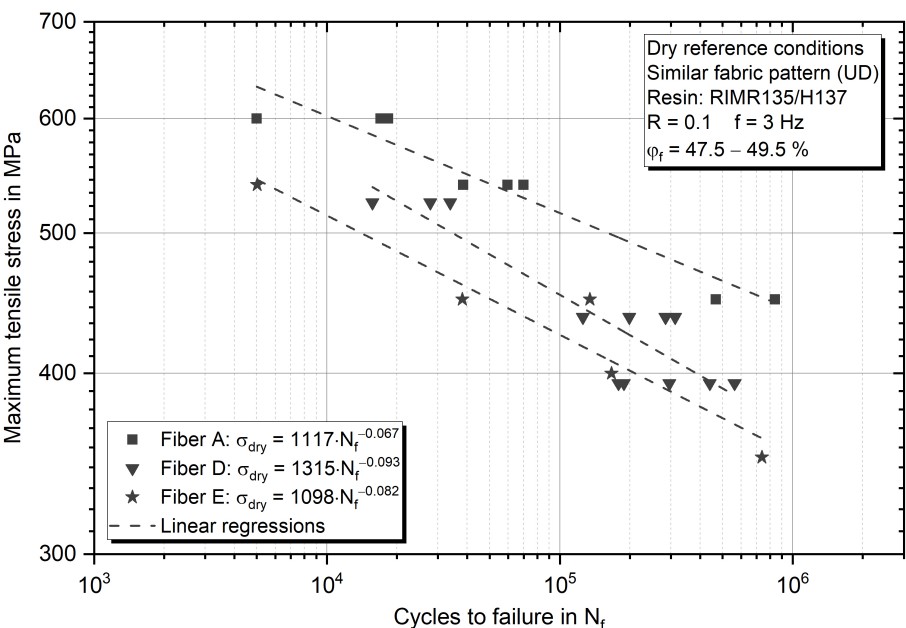

**Figure 10.** Fatigue life diagram of dry reference UD GFRP materials (0°) with initially different interphase strengths (Fiber A > Fiber D > Fiber E) and similar NCF structures.

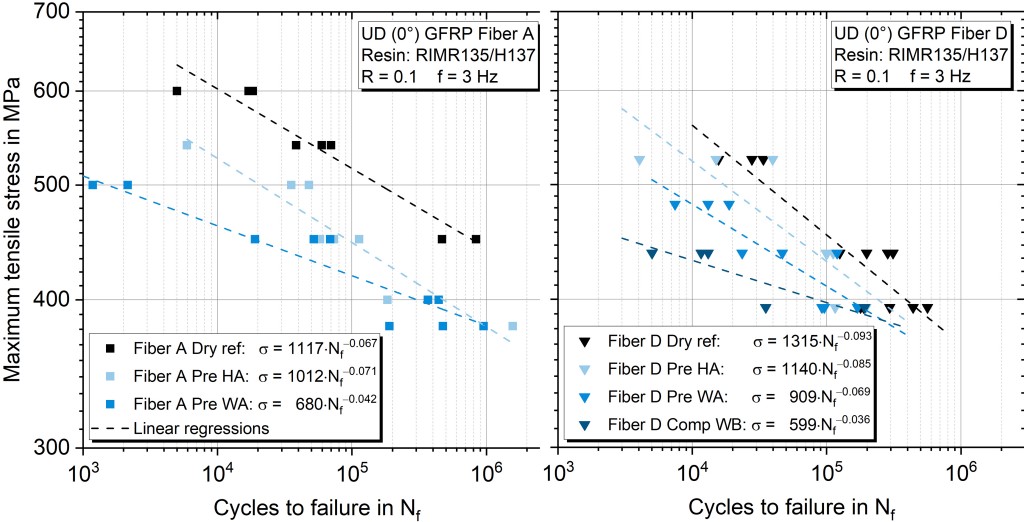

**Figure 11.** Fatigue life diagrams of fiber A (**left**) and D (**right**) UD GFRP composites (0°) in dry, humid pre-aged, wet pre-aged, and composite wet aged conditions.

The classical wet-aging of the composite in a water bath at elevated temperature further amplifies the effects described above. The slope of the Wöhler-curve drops even further and leads to a drastic reduction in the qualities at high loads. On the basis of all investigations conducted, fiber system D is expected to have a humidity- and temperature-susceptible sizing and interphase. Therefore, the fiber strength will reduce during the water bath aging of the composite, even though the epoxy matrix surrounds the fibers. For composite aging, the epoxy resin saturates with water and correspondingly changes its properties as well. Here, reductions in tensile strength, the Youngs modulus, and the glass transition temperature are the main expected differences [12,17]. On the other hand, an increasing strain to failure due to the plasticizing effect of water could also be advantageous for the fatigue performance in the high-cycle regime. Again, the lower the applied loads, the smaller the negative impact of any aging effects.

In Figure 12, representative macroscopic images of failed fatigue specimens (top) and corresponding microscopic images of broken fiber bundles inside the specimens

(bottom) are shown to analyze the different damage mechanisms. Although it is not possible to distinguish between reference specimens and pre-aged specimens based on their macroscopic appearance after testing, the microscopic view reveals major differences. In the dry reference state, bundle failure is characterized by a relatively uniform fracture pattern of fibers totally embedded and bonded with matrix resin. Thus, there is no evidence of preceding fiber/matrix debonding. In contrast, the fibers in humid pre-aged specimens are completely free of the epoxy matrix over a distance of several hundred micrometers. Here, the damage and failure mechanism has changed significantly. Though, it is expected that large fiber/matrix debondings grow after a fiber failure has occurred before the next weak fiber breaks in a larger distance than in the reference condition. For the case of wet pre-aging, the picture is different again. In fact, the fiber ends are also completely free of matrix residues, but the affected ends are significantly shorter. Considering the results of the SFF tests, this could be attributed to the drastically reduced fiber strength or the significantly increased defect density. As soon as a fiber break leads to a fiber/matrix debonding, the delamination stops early, since another weak fiber is overloaded and breaks within a narrow radius. These severe consequences of interfacial properties on the lifetime of unidirectional 0°-laminates could be revealed and characterized by means of the introduced stepwise aging methods in a new quality.

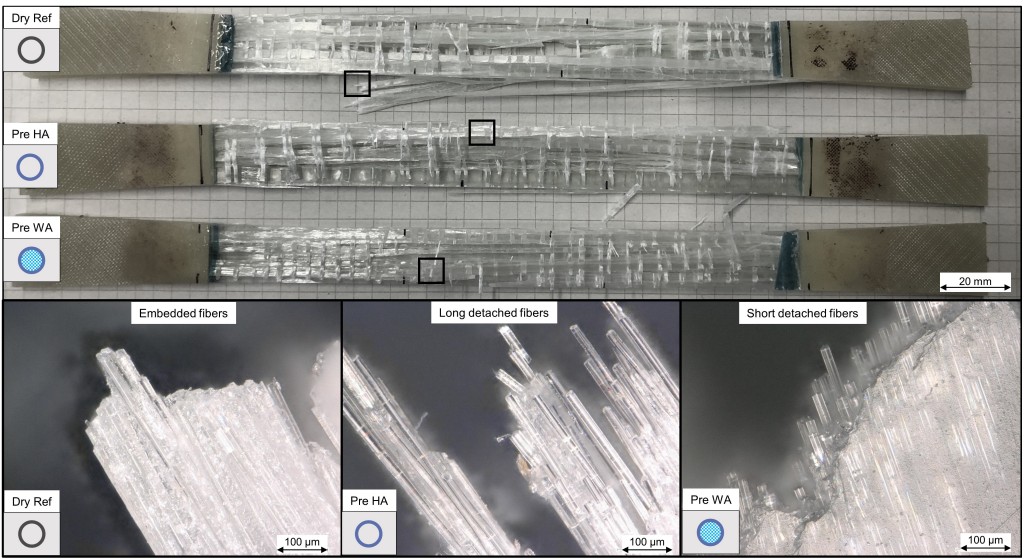

**Figure 12.** Macroscopic images of specimens after fatigue testing (**top**) and microscopic images of broken fiber bundles from specimens (black squares) in relation to different (pre)conditions (**bottom**).

## 4. Conclusions

The introduction of a new stepwise (pre-)aging of NCFs and subsequent comprehensive mechanical characterization and investigation of the lifetime properties under the influence of hygrothermal aging on five typical unidirectional GFRP composites led to several important findings. Fundamentally, it is shown that although most mechanical material properties might be similar, the selection of material is essential for fatigue properties. This applies equally to the ideal dry condition and to operating conditions influenced by humidity and temperature. The transverse tensile strength, as a measure of the fiber/matrix interphase strength, has proven to be an influential property in fatigue behavior. The interphase strength has a definite and significant impact on the performance and lifetime of unidirectional 0°-composites as it is substantially involved in the development and progression of damage.

Transportation and storage of semifinished fiber products are critical factors that can significantly limit the usability of composites. The accelerated and tailored aging process for NCFs introduced in this work allows for reliable comparisons of various systems with respect to their susceptibility to moisture-induced aging. Furthermore, for the materials

investigated so far, a correlation between good-aging resistance of the NCFs per se and good-aging resistance of the corresponding composites could also be established. However, this correlation needs to be further investigated since the water-induced chemical and physical processes in a fully formed interphase are indeed different from those on a fiber coated with sizing.

Environmental influences, have a decisive effect on mechanical properties and, in particular, fatigue lifetime. The studies in this work have shown that both the reduction in interfacial strength and the reduction in fiber strength itself result in enormous reductions in the lifetime. Aging of fiber sizing, e.g., can reduce the lifetime by up to a decade under the same loading conditions. For design purposes, the aging of sizings and the interphase, furthermore, can reduce the acceptable loads of up to 20%. Micromechanical tests and imaging confirm the significant degradation of the fiber/matrix interphase by humidity, water, and temperature. Compared to classical water bath aging, the new pre-aging methodology is suitable for revealing differences in the durability of sizings and interphases within a brief time period and without any influence from the matrix resin systems and related slow diffusion.

**Author Contributions:** Conceptualization, D.G. and C.B.; methodology, D.G.; software, D.G. and L.B.-W.; validation, D.G., C.B. and B.F.; formal analysis, D.G.; investigation, D.G. and L.B.-W.; data curation, D.G.; writing—original draft preparation, D.G.; writing—review and editing, C.B. and B.F.; visualization, D.G.; supervision, B.F.; project administration, D.G. and B.F.; funding acquisition, B.F. All authors have read and agreed to the published version of the manuscript.

**Funding:** This research received no external funding.

**Data Availability Statement:** Data will be made available on request.

**Acknowledgments:** The authors would like to thank Luc Peters and Andrey E. Krauklis for their valuable discussions on the topic.

**Conflicts of Interest:** The authors declare no conflict of interest.

## Abbreviations

The following abbreviations are used in this manuscript:

| | |
|---|---|
| FIM | Failure index matrix |
| FRP | Fiber-reinforced polymers |
| FTIR | Fourier-transform infrared spectroscopy |
| GFRP | Glass fiber-reinforced polymers |
| NCF | Non-crimp fabric |
| Pre HA | Humid pre-aging |
| Pre WA | Wet pre-aging |
| RH | Relative humidity |
| SFF | Single fiber fragmentation |
| $T_g$ | Glass transition temperature |
| UD | Unidirectional |
| WB | Water bath |

## Appendix A

**Table A1.** Test results of the quasi-static fiber bundle, longitudinal and transverse tensile tests. Strength values are given in MPa and relative to the corresponding reference strength.

| Property in MPa | Fiber A | Fiber B | Fiber C | Fiber D | Fiber E |
|---|---|---|---|---|---|
| $\sigma_{tow}$ Ref | $1138 \pm 113$ | $1300 \pm 125$ | $1123 \pm 97$ | $868 \pm 58$ | $1214 \pm 182$ |
| $\sigma_{tow}$ PreHA | $1099 \pm 144\,(-3\%)$ | $1261 \pm 74\,(-3\%)$ | $1142 \pm 45\,(+2\%)$ | $834 \pm 33\,(-4\%)$ | No data |
| $\sigma_{tow}$ PreWA | $952 \pm 77\,(-16\%)$ | $1093 \pm 40\,(-16\%)$ | $898 \pm 93\,(-20\%)$ | No data | $968 \pm 86\,(-20\%)$ |
| $\sigma_0$ Ref | $867 \pm 34$ | $991 \pm 45$ | $868 \pm 41$ | $904 \pm 21$ | $943 \pm 30$ |
| $\sigma_0$ PreHA | $904 \pm 62\,(+4\%)$ | $1095 \pm 125\,(+10\%)$ | $895 \pm 27\,(+3\%)$ | $883 \pm 25\,(-2\%)$ | $984 \pm 30\,(+4\%)$ |
| $\sigma_0$ PreWA | $681 \pm 9\,(-21\%)$ | No data | No data | $781 \pm 19\,(-14\%)$ | $828 \pm 44\,(-12\%)$ |
| $\sigma_0$ WB | $653 \pm 12\,(-25\%)$ | $771 \pm 81\,(-22\%)$ | $457 \pm 21\,(-47\%)$ | $530 \pm 20\,(-41\%)$ | No data |
| $\sigma_{90}$ Ref | $54.6 \pm 3$ | $54.8 \pm 3$ | $36.5 \pm 2$ | $47.8 \pm 3$ | $24.5 \pm 2$ |
| $\sigma_{90}$ PreHA | $40.2 \pm 2\,(-26\%)$ | $42.8 \pm 3\,(-22\%)$ | $18.4 \pm 2\,(-50\%)$ | $13.5 \pm 2\,(-72\%)$ | $19.1 \pm 1\,(-22\%)$ |
| $\sigma_{90}$ PreWA | $42.3 \pm 4\,(-22\%)$ | No data | No data | $10.2 \pm 2\,(-79\%)$ | $20.7 \pm 1\,(-16\%)$ |
| $\sigma_{90}$ WB | $46.1 \pm 2\,(-16\%)$ | $40.0 \pm 3\,(-27\%)$ | $11.6 \pm 2\,(-68\%)$ | $10.2 \pm 2\,(-79\%)$ | $12.4 \pm 3\,(-50\%)$ |

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
