# Peer review of "Influence of Sizing Aging on the Strength and Fatigue Life of Composites Using a New Test Method and Tailored Fiber Pre-Treatment: A Comprehensive Analysis"

_jcs, doi:10.3390/jcs7040139_

Round 1

Reviewer 1 Report

Comments and Suggestions for Authors

Recommendations for the Author(s):

The author reports a new method for investigating sizing and interphase-related aging effects of Composites.  

English language of the paper is excellent, the images and graphs are of a high quality.

More and more extensive applications have proved that GFRP composite is one of the most promising materials for structural lightweight. GFRP is mainly made of resin as matrix and glass fiber as reinforcement through composite process, and then becomes a reinforced material. However, in practical applications, there are still two major challenges: 1) How to avoid the damage of corrosion aging to material properties when GFRP is used in high temperature, high humidity and high salt fog marine environment; 2) When GFRP material is exposed to the above harsh environment for a period of time, how to predict its aging performance in advance and quickly. This paper attempts to solve the second challenge by providing a method of pre-aging the fiber and then re-injection resin. The ideas in the article are interesting, and preliminary effectiveness of the provided test method has also been experimentally confirmed, which makes me believe it must have practical application value. However, the description of this method is not deep enough, while both theoretical and experimental content need to be further supplemented.

1.In-depth theoretical explanation or more experiments should be provided for how the pre-treatment fibers has equivalent aging time affection comparing with normal process.

2.In line 95, the experiment was carried out for five weeks and other conditional parameters, which has a certain value. How about the results when the lasting time change? Showing more results of affection are much more convincing.

Reviewer 2 Report

Comments and Suggestions for Authors

The manuscript entiteled ‘Influence of Sizing Aging on the Strength and Fatigue Life of Composites using a new Test Method and Tailored Fiber Pre-treatment: A Comprehensive Analysis’ developed a novel fabric pre-aging method for static and fatigue testing. It enhances the analysis of the sizing, interphase, and fiber-related degradation of composites without aging them by conventional accelerated procedures or under severe maritime environments. I agree to publish this manuscript if the following questions would be elucidated.

1.Comparison with other published papers should be added, and the novelty of the new test method and tailored fiber pre-treatment should be clarified.

2.Please add a form to illustrate the Fiber A, Fiber B, Fiber C, Fiber D and Fiber E.

3.Conclusions can be concise.

Round 2

Reviewer 1 Report

Comments and Suggestions for Authors

The authors have addressed the reviewers' suggestion very well. It is good writen paper now. 

Reviewer 2 Report

Comments and Suggestions for Authors

The revised manuscript is appropriate to publish in Journal of Composites Science.